# Constructing the first comorbidity networks in companion dogs in the Dog Aging Project

Antoinette Fang[1], Lakshin Kumar[2], Kate E. Creevy[3], Daniel E.L. Promislow[4], Jing Ma[5]*
the Dog Aging Project Consortium

1 Department of Mathematics, University of Chicago, Chicago, Illinois, United States of America,
2 Department of Medicine, University of California, San Francisco, California, United States of America,
3 College of Veterinary Medicine & Biomedical Sciences, Texas A&M University, College Station, United States of America, 4 Jean Mayer USDA Human Nutrition Research Center on Aging, Tufts University, Boston, Massachusetts, United States of America, 5 Division of Public Health Sciences, Fred Hutchinson Cancer Center, Seattle, Washington, United States of America

* jingma@fredhutch.org

## Abstract

Comorbidity and its association with age are of great interest in geroscience. However, there are few model organisms that are well-suited to study comorbidities that will have high relevance to humans. In this light, we turn our attention to the companion dog. The companion dog shares many morbidities with humans. Thus, a better understanding of canine comorbidity relationships could benefit both humans and dogs. We present an analysis of canine comorbidity networks from the Dog Aging Project, a large epidemiological cohort study of companion dogs in the United States. We included owner-reported health conditions that occurred in at least 60 dogs (n = 160) and included only dogs that had at least one of those health conditions (n = 26,614). We constructed an undirected comorbidity network using a Poisson binomial test, adjusting for age, sex, sterilization status, breed background (i.e., purebred vs. mixed-breed), and weight. The comorbidity network reveals well-documented comorbidities, such as diabetes with cataracts and blindness, and hypertension with chronic kidney disease (CKD). In addition, this network also supports less well-studied comorbidity relationships, such as proteinuria with anemia. A directed comorbidity network accounting for time of reported condition onset suggests that diabetes precedes cataracts, elbow/hip dysplasia before osteoarthritis, and keratoconjunctivitis sicca before corneal ulcer, which are consistent with the canine literature. Analysis of age-stratified networks reveals that global centrality measures increase with age and are the highest in the Senior group compared to the Young Adult and Mature Adult groups. Only the Senior group identified the association between hypertension and CKD. Our results suggest that comorbidity network analysis is a promising method to enhance clinical knowledge and canine healthcare management.

**Data availability statement:** This study uses data in the Health and Life Experiences Survey (HLES) from the 2021 DAP Curated Data Release. These data are housed on the Terra platform at the Broad Institute of MIT and Harvard. Access to these data can be applied for free at https://dogagingproject.org/data-access.

**Funding:** This research was supported by the National Institutes of Health U19 grant AG057377 (DP, KC, JM), and by additional grants and private donations, including generous support from the Glenn Foundation for Medical Research, the Tiny Foundation Fund at Myriad Canada, the WoodNext Foundation, and the Dog Aging Institute (DP). DP received support from USDA cooperative agreement USDA/ARS 58-8050-9-004. The content is solely the responsibility of the authors and does not necessarily represent the official views of the National Institutes of Health. The funders had no role in study design, data collection and analysis, decision to publish, or preparation of the manuscript.

**Competing interests:** I have read the journal's policy and the authors of this manuscript have the following competing interests: DP is a paid consultant of WndrHLTH Club, Inc. and Infinity Research Labs.

## Author summary

Companion dogs age alongside humans and suffer many of the same diseases, making them an ideal "real-world" model for human health. Using owner-reported data from 26,614 dogs enrolled in the nationwide Dog Aging Project, we built the first large-scale maps—called comorbidity networks—that show which canine diseases tend to appear together and in what order. The networks correctly highlighted well-known pairings such as diabetes with cataracts and blindness, and hypertension with chronic kidney disease. They also revealed under-appreciated links—for example, protein loss in urine associated with anaemia—suggesting new avenues for veterinary research and care. By adding the reported date of diagnosis, we could infer likely sequences of the diseases: diabetes generally preceded cataracts, hip dysplasia came before osteoarthritis, and dry-eye disease often led to corneal ulcers. When we split the data by life stage, we saw disease webs become denser and more centred on a few key conditions as dogs grew older, echoing patterns seen in people. Together, these findings show that network analysis of large pet-health datasets can guide clinicians, inform breeding and prevention strategies, and ultimately improve the wellbeing of both dogs and humans.

## Introduction

Comorbidity refers to the simultaneous presence of two health conditions in an individual organism [1,2]. The understanding of comorbidities can offer insights into health condition associations and progression, which can in turn improve predictive and diagnostic tools in the clinic [3–6]. The strong positive association between measures of comorbidity and age has led those in the geroscience community to explore the mechanisms by which comorbidity might be a cause or consequence of aging [7–11]. Comorbidities are often studied in the form of a network [12–18], where each node in the network represents a health condition and each edge represents a statistically significant association between a pair of health conditions. Many health conditions are related through a variety of biological mechanisms, and comorbidity networks can help us better understand these underlying connections. Thus far, there are several large studies that use a variety of methods investigating comorbidities and comorbidity networks in humans [14,15,18–21]. The same is not true for dogs, for which there is a limited understanding of comorbidities. Notably, dogs are a good model organism because they share the same environments and many of the same health conditions as humans, they have an almost equally sophisticated healthcare system, and show similar manifestations of aging as humans [22]. Existing dog comorbidity studies are limited in sample size and diversity [23–26], or focus on comorbidities in relation to an index health condition [26–30]. While some previous studies controlled for demographic covariates when examining dog comorbidities

[26], many studies either do not report adjusting for such variables or do not explicitly describe their methods for handling dog demographics and other important covariates [23–25,27–30].

In this study, we address these limitations by constructing a comorbidity network for more than 150 canine morbidities using data from a large cohort (n = 26,614) assembled by the Dog Aging Project (DAP). The DAP is a long-term longitudinal study of the genetic, environmental and lifestyle determinants of healthy aging in companion dogs [22]. As of November 2024, more than 50,000 participants across the United States have signed up their dogs. Participating dog owners fill out an annual survey that includes extensive information on both owner demographics and dog signalment, the home and external environment, and the dog's physical activity, behavior, diet, medications, and of direct relevance to the present study, health status. Having completed this first survey, the dog becomes a member of the DAP "Pack" cohort, and all participants are invited to continue to provide additional data about their dogs over time. The data we used in this study come from the first year of DAP, and are therefore cross-sectional. Through analysis of owner-reported health data, we identify both well-documented and novel health condition associations, demonstrate age-dependent patterns in health condition co-occurrence, and investigate temporal relationships between health conditions. These findings contribute to veterinary informatics and health policy, expanding opportunities for evidence-based medicine in veterinary practice.

## Results

### Study cohort

We included health conditions that occurred in at least 60 dogs, and we only included dogs that had at least one of these conditions. The final cohort encompasses 160 unique health conditions and 26,614 dogs, each with at least one of the 160 conditions. Sex and reproductive status were relatively balanced, with spayed females comprising 12,457 (46.81%) of the sample, and neutered males accounting for 12,196 (45.83%). Intact females (687, 2.58%) and intact males (1,274, 4.79%) were less common, with only 7.37% of dogs in the cohort being reproductively intact overall. The breed distribution was approximately equal, with 13,407 (50.38%) mixed breed dogs and 13,207 (49.62%) purebred dogs. The median age of dogs in the study was 7.75 (interquartile range [IQR]: 4.08-11.00 years), and the median weight was 50.90 lbs (IQR: 25.00-69.0 lbs [11.34-31.30 kg]). The health conditions are grouped into 20 body system categories. The skin category contains 21 health conditions (the highest number), while some other categories (congenital, immune-mediated, and hematopoietic) contain only one health condition (Sheet A in S1 Table). Additionally, 28.4% of dogs have two or fewer conditions, while a substantial 71.6% exhibit multimorbidity with three or more concurrent health conditions (Fig 1B). The most frequently observed condition is fractured bone, followed by dental calculus, extracted teeth, and dog bite (Fig 1C). The difference between each dog's most recent date of reported condition onset and earliest date of reported condition onset was used as a proxy for its medical history span (Fig 1A). Unsurprisingly, dogs' medical history increases as they age, except for those over 21 years old. Knowing the span of each dog's medical history allows us to study directed comorbidities across prespecified time windows.

### Association between demographic factors and health conditions

Over 80% of the models in which the age coefficient was statistically significant (P < 0.05) had a positive age coefficient (Fig 2). The age effect is particularly strong for conditions in the bone/orthopedic, brain/neurologic, cancer, cardiac, and liver/pancreas categories. Conditions that had a significant negative coefficient for age were conditions we would expect to occur in younger dogs, such as retained baby teeth, cryptorchidism, or coccidia [31]. Weight (as a measure of body size) showed significant associations with approximately 70% of health conditions, with 58% of these significant associations being positive. All cancer types showed significant positive associations for age (P < 0.001) and weight (P < 0.05). The sex variable coefficients were not statistically significant for over 90% of the conditions. There were several health conditions for which correlation coefficients differed by sex. However, these health conditions also tend to be those that affect the reproductive organs of male or female dogs (e.g., cryptorchidism or pyometra) [31]. The breed background coefficient

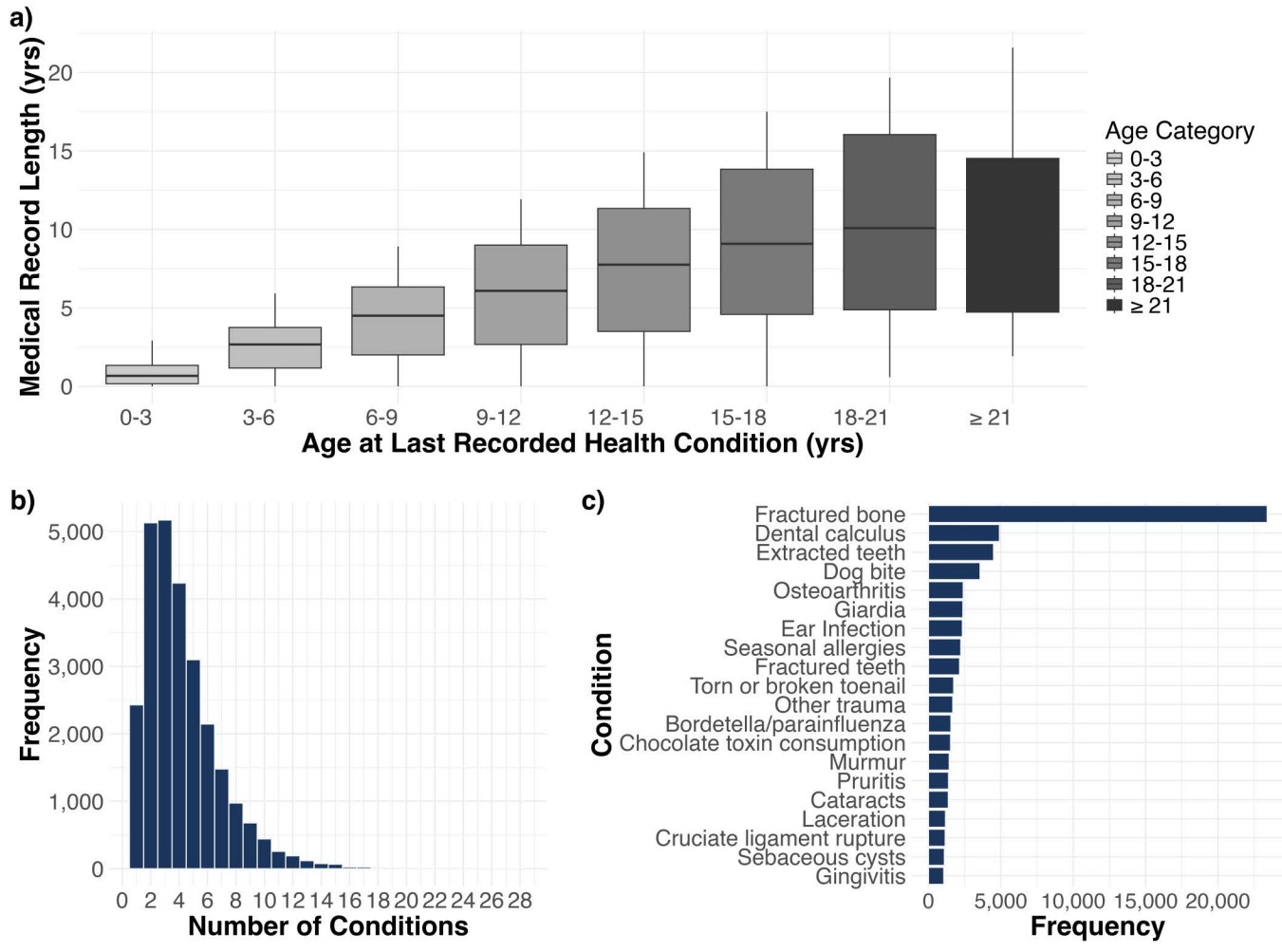

**Fig 1. Summary of health condition distribution and medical history characteristics. (a)** Boxplot comparing each dog's age at last reported health condition with the total span of their medical history (i.e., time between first and last reported condition). **(b)** Histogram representing number of dogs with different numbers of conditions. **(c)** Bar chart displaying the 20 most frequently occurring health conditions in the study population.

was only significant for around 47% of the models, and of those, less than 40% were positive (i.e., greater in mixed breed dogs). Purebred dogs showed significantly higher rates of ear infection, cataracts, and intervertebral disk disease (IVDD). In comparison, mixed-breed dogs had significantly higher rates of cruciate ligament rupture, being hit by a car, and many conditions in the infection/parasites category. No significant differences were found between purebred and mixed breed dogs for conditions such as hip dysplasia, patellar luxation, hypoadrenocorticism, hyperadrenocorticism, and mast cell tumors.

## Undirected comorbidity network

The undirected comorbidity network is presented in Fig 3. A complete table of all health condition pairs and their corresponding P-values, regardless of statistical significance, is provided in Sheet A in S2 Table. Of note, health conditions of the same categories tend to be connected with each other. This network highlights a variety of well-known comorbidities: diabetes and blindness (P<0.001); diabetes and cataracts (P<0.001); cataracts and blindness (P<0.001); Cushing's

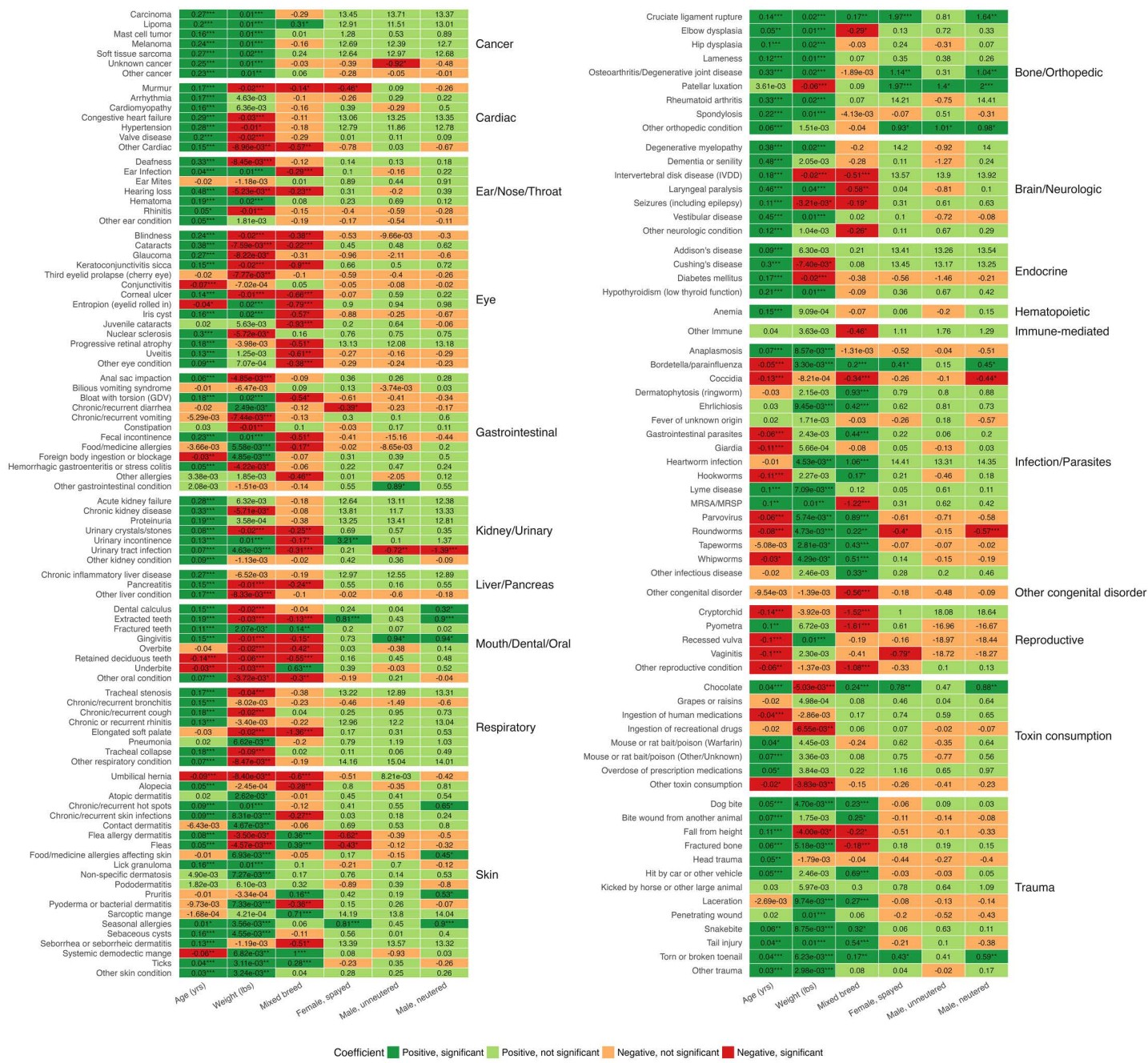

**Fig 2. Associations between demographic factors and health conditions, stratified by condition category.** Coefficients from logistic regression models are shown for each health condition, with age, weight, breed background, and sex/reproductive status as predictors. Statistical significance is denoted by asterisks (* p < 0.05, ** p < 0.01, *** p < 0.001).

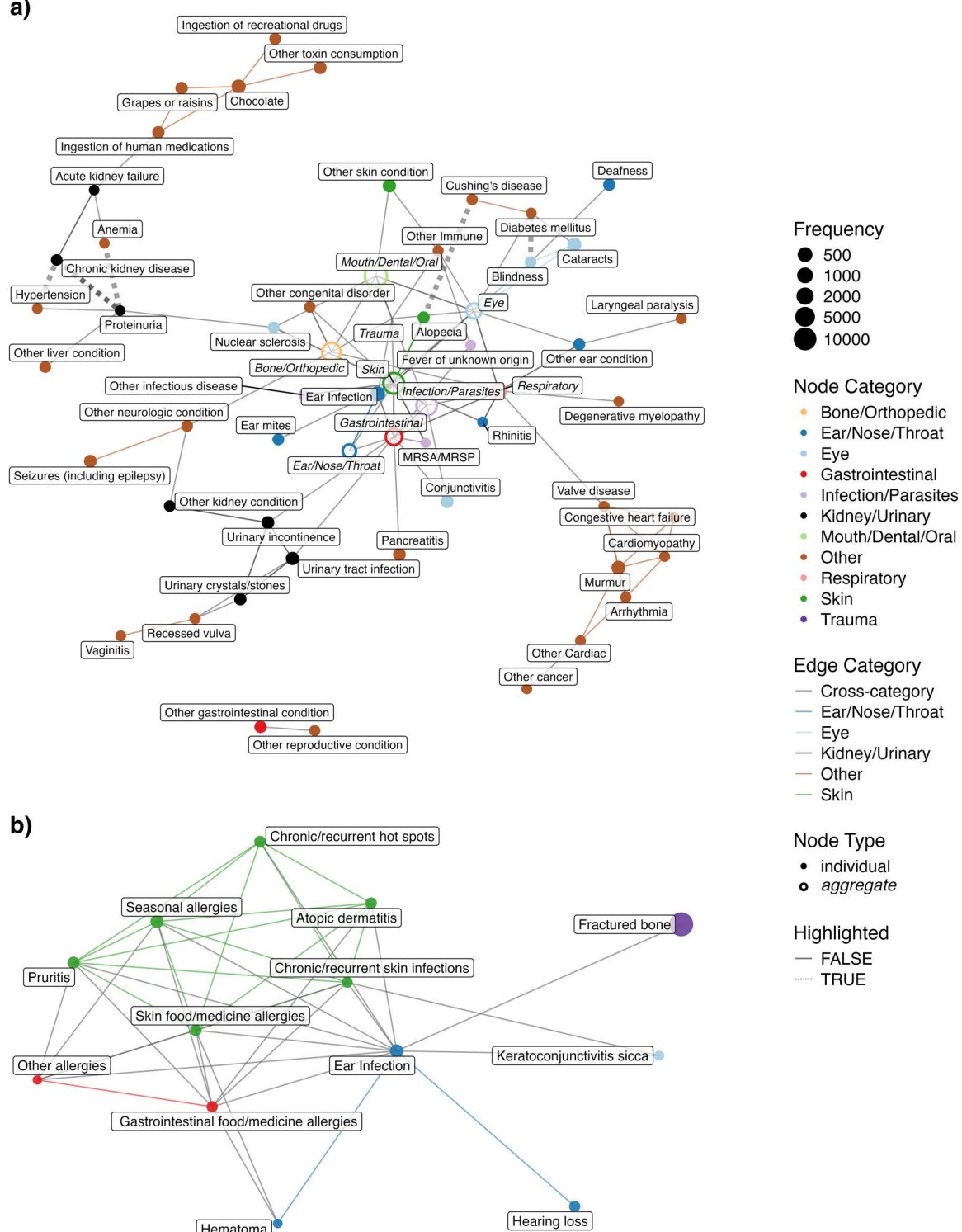

**Fig 3. Undirected comorbidity network.** Nodes are health conditions with node size proportional to condition prevalence, and edges are statistically significant comorbidity connections (at a Bonferroni adjusted P-value < 0.001). Colors indicate health condition categories. **(a)** Full network view with Bone/Orthopedic, Ear/Nose/Throat, Eye, Gastrointestinal, Infection/Parasites, Mouth/Dental/Oral, Skin, and Respiratory categories collapsed into single aggregate nodes. **(b)** Zoomed-in view of the Skin conditions, including their first-degree neighbors to ear infection.

disease and alopecia (P<0.001); and atopic dermatitis and various allergies— specifically skin food/medicine allergies (P<0.001), seasonal allergies (P<0.001), and gastrointestinal food/medicine allergies (P<0.001). All of these pairs include eye or skin health conditions, and these health conditions are commonly used by veterinarians as indicators of systemic diseases like diabetes [31,32]. In addition to these skin- and eye-related health condition pairs, we also see some other health condition pairs like hypertension and chronic kidney disease (CKD) (P<0.001); hypertension and proteinuria (P<0.001); and proteinuria and anemia (P<0.001). The network also demonstrates the central role of ear infections with a high degree of connectivity (11 connections), linking it to multiple dermatological conditions like atopic dermatitis or chronic and recurrent skin infections; various forms of allergies like food/medicine allergies affecting the skin or seasonal allergies; and other complications like hearing loss and keratoconjunctivitis sicca. Lastly, the network reveals a notable cluster of parasite and parasite-related health conditions, which supports the observation that parasites often co-occur [33–35].

Next, we examined the node degree distribution to see if the network can be classified as random or scale-free. Node degree, which is defined as the number of edges connected to each node [36], is one measure of node centrality. The degree distribution is one of the most fundamental properties of networks and provides important information about a network's structure [37]. The node degree distribution of our network is better fitted with an exponential distribution than a scale-free power-law distribution (P<0.001) (S1 Fig), providing evidence for a random network.

## Comparison to stratified networks

We stratified the dataset into four groups by life stage (Puppy: n=476; Young Adult: n=3931; Mature Adult: n=15895; Senior: n=6155) to investigate the extent to which comorbidity networks varied across life stages (see Methods). As expected, the four networks constructed using the stratified approach differed greatly from each other and also from the unstratified network (Fig 4). Notably, no significant comorbidity patterns were detected for the puppy age group, which comprised less than 2% of the cohort. Puppies had only three conditions meeting the inclusion threshold and a much lower proportion of dogs with two concurrent conditions (28%) compared to other age groups. The young adult, mature adult, and senior networks have, respectively, 19, 97, and 57 conditions. A complete table of all health condition pairs and their corresponding P-values in these 3 networks, regardless of statistical significance, is provided in Sheets B, C, and D in S2 Table. The mature adult network has the most number of edges with a large connected component (Fig 4B). In all three networks, conditions of the same type tend to be connected with each other, an observation also seen in the unstratified network.

Thirteen comorbidity pairs were identified across all four networks: extracted teeth – fractured teeth, skin food/medicine allergies – gastrointestinal food/medicine allergies, chronic or recurrent diarrhea – chronic or recurrent vomiting, chronic or recurrent diarrhea – gastrointestinal food/medicine allergies, hookworms – roundworms, roundworms – tapeworms, atopic dermatitis – pruritis, pruritis – seasonal allergies, fleas – ticks, skin food/medicine allergies – seasonal allergies, seasonal allergies – gastrointestinal food/medicine allergies, atopic dermatitis – skin food/medicine allergies, and chocolate consumption – grapes/raisins consumption. Conversely, there were 12 unique comorbidity pairs identified by the stratified networks but not by the unstratified network. Eleven of these pairs came from the mature adult group and one from the senior group (Sheets A and B in S3 Table). The pair unique to the senior group is between ticks and gastrointestinal food/medicine allergies. Many pairs unique to the mature adult group are expected, such as elbow dysplasia/ lameness/ spondylosis – osteoarthritis, murmur/ cardiomyopathy – chronic or recurrent cough, and contact dermatitis – flea-allergy dermatitis. However, there is also one pair between other respiratory conditions and other immune conditions, and pairs that are not intuitive, e.g., Conjunctivitis – Giardia and cataracts – deafness. On the other hand, the unstratified network identified 110 comorbidities that were not detected by *any* of the stratified networks. However, this could be due to lower statistical power in the stratified analysis. Among the stratified networks, only the senior network identified the association between hypertension and CKD, while only the mature adult network identified the comorbid pair of blindness and cataracts.

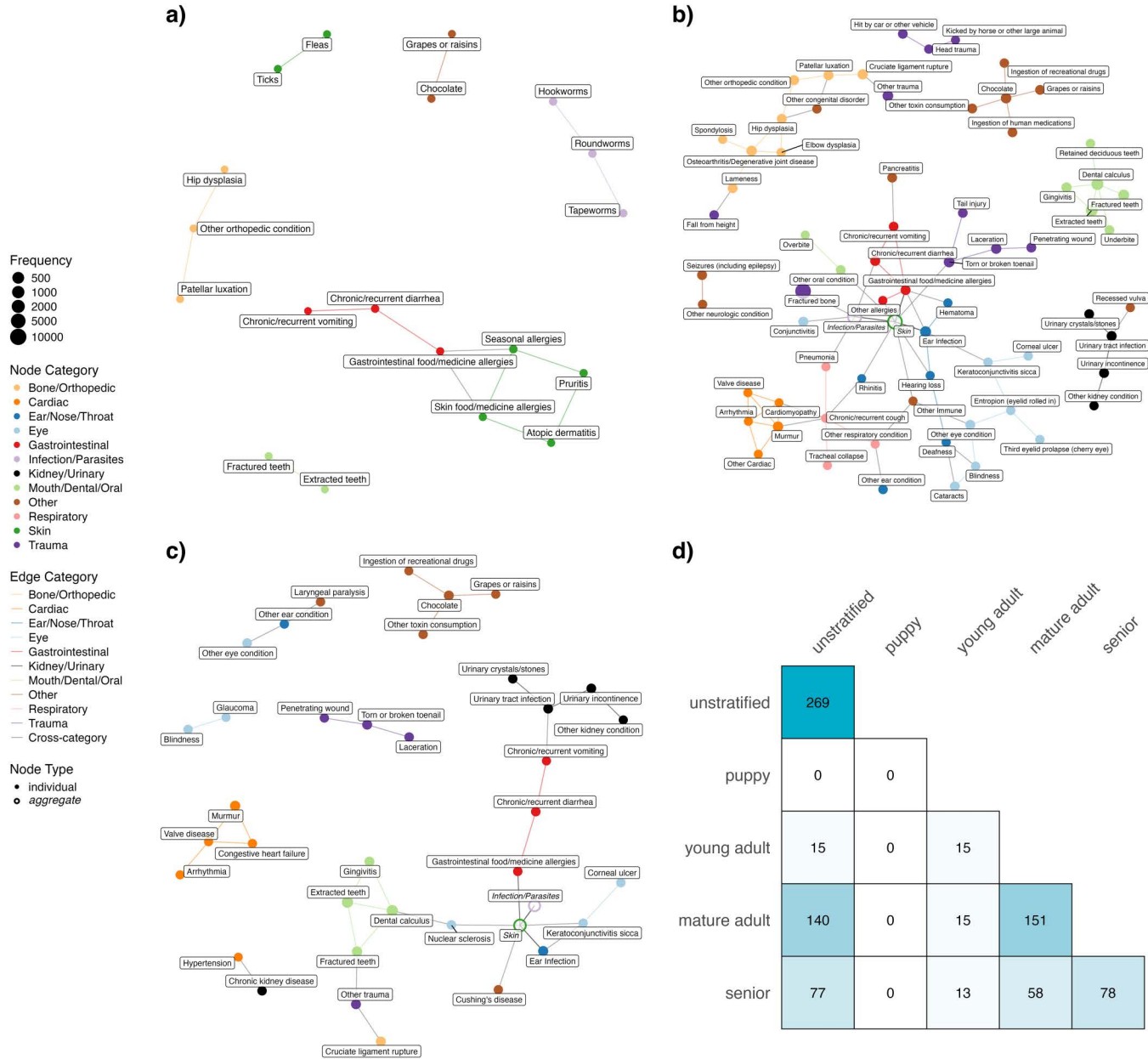

**Fig 4. Age-stratified comorbidity networks.** Undirected comorbidity networks in Young adult **(a)**, Mature adult **(b)** and Senior **(c)**. In each network, nodes represent health conditions with node size proportional to condition prevalence, and edges represent statistically significant comorbidity associations (at a Bonferroni-adjusted P-value < 0.01). Colors indicate health condition categories. No network is shown for the Puppy stratum as no significant comorbidities were detected in this age group. Conditions in the Skin and Infection/Parasites categories are presented as aggregate nodes in (b) and (c) to minimize cluttering. **(d)** Heatmap displaying edge overlap between disease networks across age groups. Numbers indicate shared edges between a pair of networks, with diagonal values showing total number of edges per network.

Network topology analysis of the age-specific networks revealed distinct patterns in disease interconnectedness (Table 1). Edge density displayed a different trajectory than other measures, where it was the highest in the young adult network. By contrast, all other measures– clustering coefficient, betweenness centrality, and degree centrality– increased with age.

**Table 1. Network topology measures across life stages:** edge density refers to the number of edges relative to the maximum number of possible edges in each graph; clustering coefficient measures the degree to which nodes in each graph cluster together; betweenness and degree centrality refer to the normalized graph level centralization measure based on the respective node-level centrality scores.

| | Puppy | Young Adult | Mature Adult | Senior |
|---|---|---|---|---|
| **Edge Density** | N/A | 0.088 | 0.034 | 0.046 |
| **Clustering Coefficient** | N/A | 0.214 | 0.491 | 0.480 |
| **Betweenness Centrality** | N/A | 0.047 | 0.133 | 0.149 |
| **Degree Centrality** | N/A | 0.088 | 0.083 | 0.113 |

This suggests a progression from a less structured and more diffuse network in the young adult life stage to an increasingly modular network with a few hub nodes as dogs age. Together, these results support the idea that age is a critical factor influencing comorbidity development.

### Adding temporal P-values to understand health condition progression

Finally, we examined temporal relationships between conditions by computing the probability of one condition preceding another within a time window of 12 months and testing its significance using the Poisson binomial approach to comorbidity (see Methods for details). Shorter window spans are more adept at capturing comorbidities that occur sequentially [18], but could result in power loss in comorbidity discovery since shorter window sizes encompass fewer dogs. Additionally, in shorter time windows, condition development, appearance of relevant signs and symptoms, provider diagnosis, and the recording of said diagnosis may not be accurately, meaningfully captured. The ideal window size should be guided by clinical knowledge and research objectives [18]. In our case, we chose a 12-month window to balance capturing meaningful temporal associations while minimizing potential confounding by intervening events or treatments.

A time-directed comorbidity network is presented in Fig 5. A complete table of all health condition pairs and their corresponding P-values in these 3 networks, regardless of statistical significance, is provided in Sheet E in S2 Table. Similar to the undirected network case, conditions of the same category tend to be connected with each other with a dense cluster of skin-related conditions. The directed network suggests that diabetes occurs before cataracts (P < 0.001), keratoconjunctivitis sicca (KCS) before corneal ulcer, elbow/ hip dysplasia before osteoarthritis, hypertension before CKD, pneumonia before bordetella/parainfluenza, congestive heart failure before valve disease, and blindness before cataracts. A full list of directional pairings supported by existing literature can be found in S4 Table.

## Discussion

Aging and comorbidities are linked in a self-reinforcing feedback loop [8]. Biological aging leads to increased risk of various diseases, which increases the risk of comorbidities. The presence of these comorbidities contributes to further physiological decline which accelerates biological aging. The dog is a great model to explore this relationship. Through the use of canine comorbidity networks, this study represents an important step towards understanding the relationship between aging and comorbidity. Our work also lays a critical foundation for incorporating future DAP genetic and other molecular data to elucidate the mechanisms underlying the association between aging and comorbidity.

Our logistic regression analyses provide a comprehensive overview of demographic and biological factors associated with the occurrence of various canine health conditions. Our results reaffirm that age is the primary risk factor for many health conditions, while body size and breed background modulate disease risk in condition-specific manners. In our models, 80% of the coefficients in which age is significant are positive, confirming that morbidity rises steeply with older age. Other large epidemiological studies support this pattern [38,39]. Weight is significant in about 70% of the conditions, with 57% positive associations. Separate investigation using DAP data agreed that larger dogs had higher lifetime prevalence of cancer and orthopedic disorders but lower odds of ocular, cardiac, hepatic and

**Fig 5. Time-directed network using a window size of 12 months.** Nodes are health conditions, and edges are statistically significant comorbidity connections (at a Bonferroni-adjusted P-value < 0.01). Colors indicate health condition categories. Arrowheads point from the health condition that occurs earlier in time to the health condition that occurs later. **(a)** Full network view with Skin and Infection/Parasites categories collapsed into single aggregate nodes. **(b)** Zoomed-in view of the Skin and Infection/Parasites conditions, including their first-degree neighbors.

respiratory disease [40]. For cancer specifically, heavier dogs were diagnosed several years earlier than lighter dogs [38]. These external findings corroborate our weight coefficients and clarify that "size risk" is condition-specific rather than universally detrimental. Our models did not consider the interaction between age and body size, which may be included to further refine disease risks [40]. The effects of breed background are highly condition-specific and not uniformly disadvantageous to purebreds. Consistent with our findings, a study based on the UK VetCompass cohort showed that ear infections were more common in purebreds [39]. A US study based on electronic health records of 27k dogs found purebreds over-represented in cataracts and IVDD, but no difference in hip dysplasia, patellar luxation, hypoadrenocorticism, hyperadrenocorticism, and mast cell tumors, while mixed-breed dogs had excess cruciate-ligament rupture and being hit by a car [41], which are aligned with our discoveries. On the other hand, their results on elbow dysplasia (bone/orthopedic) and hypothyroidism (endocrine), both over-represented in purebreds, are distinct from ours. This could be due to our use of a multivariate logistic regression model, where the effect of breed background is after adjusting for other covariates. We found that mixed-breed dogs had higher rates of infectious and parasitic diseases. This pattern does not appear to stem from poorer husbandry: mixed-breed owners actually reported flea- and tick-control use more often than purebred owners and had comparable rates of heartworm prevention. Although mixed-breed owners were more likely to be in lower-income brackets, income alone therefore does not explain the difference. A plausible alternative is acquisition history: mixed-breed dogs are more often adopted as adults—particularly from shelters—where they may have contracted infections or parasites before joining their current households. Lastly, it is also worth noting that breed background is very imprecise. Future work should incorporate genetic data to fully disentangle the effect of breed on disease risk.

The undirected canine comorbidity network supports clinical observations that ear infections often occur as part of a broader pattern of allergic and inflammatory conditions, particularly those affecting the skin [42]. The multiple connections to allergic conditions align with previous findings that hypersensitivity diseases are among the most common primary factors leading to otitis in dogs [42]. The connection between Cushing's disease and alopecia is well-known: the disease leads to overproduction of the hormone cortisol, which affects hair growth and can cause the hair to fall out [43]. The conditions diabetes, blindness and cataracts are associated with each other in our study, which are aligned with clinical observations that most dogs with diabetes mellitus will acquire cataracts in 1–2 years [44]. In addition, the network highlights the association between anemia, proteinuria and CKD. Proteinuria describes the presence of protein in the urine and may be associated with acute or chronic kidney disease, as well as with certain systemic health conditions [40]. Anemia, a deficiency of red blood cells, is also often present in advanced CKD due to the combination of impaired red blood cell survival deficiency of production of erythropoietin by the kidney, and impaired bone marrow responsiveness to erythropoietin [45–47]. The network illustrates these relationships by connecting anemia to CKD through proteinuria. While our undirected network cannot imply causal relationships or mediation effects, it does highlight potential associations that warrant further investigation. A recent study conducted with a small cohort of 37 dogs suggests that proteinuria may contribute to anemia in dogs with CKD, a relationship that has also been suggested in other model organisms [48–50]. The connections revealed by our network demonstrate its potential in identifying less obvious health condition associations and comorbidities. The network's degree distribution is better modeled with an exponential distribution when compared with a scale-free power-law distribution. Had we observed support for the power-law distribution, this could have been interpreted as suggesting that the risk of having an additional morbidity is directly proportional to the number of morbidities a dog already has. By contrast, the exponential distribution is consistent with a model where the probability of developing a morbidity is independent of the number of morbidities a dog already has. However, we note that since this is a small network, observing a power-law degree distribution was unlikely to begin with [51,52].

Analysis stratified by life stage revealed important age-specific patterns in canine comorbidities. The mature adult network includes more conditions and edges than the young adult and senior networks due to the larger sample sizes, but the senior network is slightly more complex in terms of edge density and centrality measures. Interestingly, only the senior

network identified the comorbid pair of hypertension and CKD. Both diseases, as well as the condition categories they belong to (cardiac and kidney/urinary) are of particular interest in aging research.

When comparing the unstratified network to the age-stratified versions, the unstratified network revealed numerous associations that were absent in the stratified analyses—likely due to reduced statistical power from smaller sample sizes. Conversely, the age-stratified networks uncovered several associations that did not appear in the overall network, highlighting patterns that may be specific to particular life stages. The ticks – gastrointestinal food/medicine allergies pair is unique to the senior network. Neither condition is strongly associated with age, yet their comorbidity is only represented in the senior age group. The association between tick bites and alpha-gal syndrome (AGS) is well-documented in humans: the lone star tick is the primary source in transmitting the alpha-gal sugar that triggers AGS—an allergy to red meats and other mammalian products [53]. In dogs, tick bites can also induce alpha-gal antibodies, but no clinical signs of food allergy were observed [54]. Further investigation is warranted to understand this comorbidity pair. The conjunctivitis – giardia pair unique to the mature adult network is unexpected. Given that both conditions involve mucosal inflammation, it could be that one or more genetic factor(s) make dogs susceptible to both conditions. Similarly, there may be a genetic component underlying the association between cataracts and deafness. For example, breeds carrying the merle allele (Australian Shepherd, Collie, Shetland Sheepdog, Great Dane, Dachshund, Catahoula Leopard Dog, etc.) have strict breeding recommendations that avoid mating two merles, preventing the high-risk M/M genotype that is associated with sensorineural deafness and ocular anomalies [55]. Most of the common comorbidities identified by all networks regardless of stratification involve skin and/or gastrointestinal conditions that tend to have weaker—or even negative—associations with age than do cancer, endocrine, cardiac or brain/neurologic conditions. This suggests that while the unstratified network is valuable for broad discovery, combining it with stratification can provide further insights when focusing on specific conditions or specific groups of health conditions.

Finally, incorporating the time of reported condition onset provides us with some insights into health condition progression, which could be beneficial for canine health management. Our results show that diabetes occurs before cataracts, which is consistent with the literature: it is estimated that around 75% of dogs diagnosed with diabetes will have blinding cataracts within the following two years [44]. KCS, also known as dry eye, is a frequent canine ophthalmic disease and can lead to corneal ulcers in dogs [56]. Osteoarthritis is a disease involving the entire joint organ and all of its supporting tissues, though it is most commonly marked by the degeneration and dysfunction of articular cartilage [57]. Joint dysplasia—typically affecting the hip or elbow—results from abnormal joint development and can advance to osteoarthritis [58]. Our network identifies this causal cascade by linking elbow dysplasia to hip dysplasia and finally osteoarthritis. Moreover, our network reveals a directional pairing from hypertension to CKD. This not only reaffirms the association identified in the undirected network, but also corroborates with existing literature which found systemic hypertension causes renal injury to dogs [59]. On the other hand, the time-directed network also reveals several counter-intuitive pairings: congestive heart failure→valve disease, pneumonia→bordetella/parainfluenza, and blindness→cataracts. One possible explanation for all three cases is reporting lag: owners may be informed of one condition and learn the specific cause later, reversing the apparent time-stamp. In the case of pneumonia→bordetella/parainfluenza, some dogs receive the bordetella/parainfluenza vaccine during hospital recovery, creating a false post-pneumonia "exposure" entry.

This study has several limitations. First, health condition data were based on owner-reported lifetime prevalence collected at a single point, which introduces recall bias. Owners may not accurately recall all of the dog's health conditions or misremember the order in which the conditions occurred. Furthermore, the cross-sectional design is subject to survivor bias: only dogs who have survived to the point of entry into the study are included. This could distort the observed comorbidity patterns, especially for conditions like cancer that are associated with increased mortality, as dogs that died from these conditions prior to recruitment are not represented in our sample. As DAP collects longitudinal data, these issues may be partially addressed, but they remain a limitation of the current analysis. Second, owners may misreport health status information. The owner-reported dates of condition onset do not necessarily represent the pathophysiological course

of a health condition, or may themselves be inaccurate. The survey was designed to be user-friendly, but some of the health conditions overlap or are medically imprecise. Also, health condition categories better align with organ systems, so a single pathological process that is implicated in multiple systems could end up being described in the survey as two or more health conditions. We did not analyze the free-text responses provided by participants in the "other, please describe" fields. These free-text fields were used by about 10% respondents and may contain valuable information. Future work will involve the systematic categorization and analysis of these responses. Third, veterinarians may only recommend, or owners may only elect to proceed with, certain diagnostic tests based on prior knowledge of a dog's current diagnoses, which could introduce confirmation bias into the network. For instance, blood pressure testing is likely to be recommended in dogs with diagnosed kidney disease, due to the known association of hypertension with kidney disease. However, dogs without kidney disease may not be tested, so lack of an owner-reported condition does not imply its true absence. Furthermore, dogs in DAP might be healthier than average if owners are disinclined to sign up a very unhealthy dog for a study on the determinants of healthy aging, similar to the well-known "healthy volunteer" effect in human studies [60–62]. Fourth, breed classification (purebred vs. mixed-breed) is also owner-reported. While this binary classification eases statistical analysis, we cannot verify the accuracy of breed designations without genetic testing, which is only available for a subset of dogs in our cohort. Fifth, rare health conditions pose challenges for logistic regression models, increasing the risk of overfitting and potentially misrepresenting the relationship between rare and common health conditions. For example, although allergies are a common cause of pododermatitis [63], this relationship was not detected in our network—likely due to the small number of dogs with pododermatitis (n = 65) compared to allergies (n = 1015). Finally, our study cohort is not demographically representative. The owners in our study are predominantly white, highly educated, and wealthier than the US population (S5 Table), which may limit the generalizability of our findings. Additionally, we only considered biological factors (age, sex, sterilization status, breed background, and weight) in disease risk prediction. Environmental and social factors can also affect disease risk [64] and should be incorporated in future analyses.

Acknowledging these limitations, this study demonstrates the power of comorbidity discovery from DAP epidemiological surveys, which has great potential to uncover further insights. First, while this report focuses on positively correlated health condition pairs, our analysis also identified several negatively correlated pairs. A systematic investigation of these negative associations could reveal potential antagonistic relationships between conditions, though careful interpretation would be needed to distinguish true biological relationships from competing risk effects. Second, DAP has access to veterinary electronic medical records (VEMRs) that can be mined using natural language processing and analyzed using the network methods. It would be interesting to compare a network learned from VEMRs with the one inferred from owner-reported data. Furthermore, while our analysis focuses primarily on pairwise disease associations (comorbidities), many dogs in our cohort (71.6%) exhibit multimorbidity with three or more conditions. Future work could extend these network approaches to better understand complex interactions between multiple diseases simultaneously. This direction would align with the growing interest in multimorbidity research in human medicine. Lastly, we have already collected genetic profiles for over 7000 dogs in DAP. Future work could use gene-relatedness learned from these genetic profiles to control for genetic differences among dogs, potentially surpassing the precision of our current binary classification of breed background. The inclusion of genetic profiles would better reflect the subtle but important differences between various dog breeds and help elucidate the molecular mechanism underlying canine comorbidities.

## Methods

### Ethics statement

The University of Washington IRB deemed that recruitment of dog owners for the Dog Aging Project, and the administration and content of the DAP Health and Life Experience Survey (HLES), are human subjects research that qualifies for Category 2 exempt status (IRB ID no. 5988, effective 10/30/2018). No interactions between researchers and privately owned dogs occurred; therefore, IACUC oversight was not required. The studies were conducted in accordance with the

local legislation and institutional requirements. The participants provided their written informed consent to participate in this study.

### Dog Aging Project owner survey data

In the Dog Aging Project, data on signalment, husbandry and health status of the dogs in the study are collected from all participants through a comprehensive owner-reported survey; subsets of participants provide other types of information as well. The Health section of the baseline Health and Life Experiences Survey (HLES) captures lifetime prevalence of health conditions, defined as the proportion of individuals in a population who have experienced a particular condition or event at any point in their lives up to the time of the survey or study. Owners follow survey logic to identify their dogs' health conditions or clinical signs, organized by process (e.g., trauma) or body system (e.g., cardiac). Owners are presented with more than 300 different health conditions or clinical signs, with an option for "other, please describe" in each body system, and also provide the time of onset for each reported condition. This study utilized the 2021 DAP Curated Data Release, which contains all survey responses obtained by 12/31/2021. As such, while the data we analyze are cross-sectional because they were reported at a single point in time, the survey items are constructed to capture lifetime prevalence as defined above.

Of note, the following information from this dataset was used: [1] age, sex, sterilization status, breed background, and weight of each dog, and [2] list of health conditions (out of 365 possible, distinct health conditions, Sheet B in S1 Table) and the date of reported condition onset (if any) for each dog. Note that the list of health conditions here reflects the health condition options presented to dog owners on the survey.

### Logistic regression modeling

To estimate the individualized probabilities for each health condition, we fit logistic regression models (LRM). Previous veterinary work suggests that factors such as age, sex, spay/neuter status, weight, and breed background are associated with health condition development in dogs [39,40,45–47,65]. Thus, each LRM included the following demographic characteristics as covariates: age, weight, sex, and whether the dog is purebred or mixed-breed. Sex is defined as a four-category variable (intact female, spayed female, intact male, neutered male). Thus, there are in total six predictors in each LRM. We used breed background as opposed to breeds as a covariate because there are a large number of breeds and many health conditions are very rare. Fitting LRMs on rare events data with a large number of covariates can lead to large bias in estimation and statistical inference [66]. Despite being an imprecise summary, breed background has previously been reported to be associated with several health conditions [46]. The response variable is binary indicating whether the dog has a given health condition. The LRMs estimate each dog's probability for having a certain health condition $P_{y \in d}$ :

$$P_{y \in d} = \frac{1}{1 + exp(-(\beta_0 + \beta_1 x_1 + \ldots + \beta_6 x_6))}.$$

(1)

To minimize the risk of overfitting, we applied the one-in-ten rule of thumb for logistic regression analysis, which suggests studying one predictive variable for every ten events [67]. Thus, we only included health conditions that occurred in at least 60 dogs, given the six covariates in our model.

### Comorbidity P-values

The null hypothesis is that a pair of health conditions $y, z$ are independently distributed: $H_0 = y \perp z$. A significant P-value indicates evidence that the health condition pair co-occur more often than by chance. Previous canine comorbidity literature largely relies on methods that assume that the probability of seeing health conditions $y, z$ in a dog $P_{\{y,z\} \in d}$ is equal to the product of the population incidence rate for each health condition under the null hypothesis:

$$P_{y \in d} = \frac{C_y}{N}, \; P_{z \in d} = \frac{C_z}{N}$$

(2)

$$P_{\{y,z\} \subset d} = \frac{C_y}{N} \times \frac{C_z}{N}$$

(3)

where $C_y$, $C_z$ represents the prevalence of health conditions and $N$ the total number of dogs. A comorbidity P-value can then be calculated under the binomial distribution. However, this method does not account for important covariates that could impact individual disease risks. A standard approach to deal with covariates is to stratify the population into homogeneous subgroups [15,18,19]. The downside of this approach is that it greatly decreases statistical power [18]. To overcome this issue, we used the Poisson binomial approach to comorbidity (PBC) [18] method to calculate P-values for each health condition pair. We opt for the PBC method as opposed to other notions of health condition associations [16], because PBC does not require sample stratification and has been shown to have higher power in comorbidity discovery [18]. The probability that two health conditions occur by chance in a dog is simply the product of the two individualized health condition probabilities given by the LRMs described previously:

$$P_{\{y,z\} \subset d} = P_{y \in d} \times P_{z \in d.}$$

(4)

In short, the Poisson binomial distribution is a generalization of the binomial distribution in which every dog has its own probability of having a health condition. Since the cumulative distribution function for the Poisson binomial is computationally unwieldy, we used the normal approximation, which is accurate but also faster [18,68]. To determine a P-value approximately, we estimated the mean and variance under the normal distribution as follows. The mean in this case represents the expected number of dogs with both health conditions:

$$\mu_{y,z} = \sum_d P_{\{y,z\} \subset d.}$$

(5)

The variance is

$$\sigma_{y,z}^2 = \sum_d \sigma_{P_{\{y,z\} \in d}}^2 = \sum_d P_{\{y,z\} \subset d} \left(1 - P_{\{y,z\} \subset d}\right).$$

(6)

Initial analyses showed that accounting for the uncertainties in the predicted individual probabilities through the delta method did not meaningfully alter the comorbidity relationships identified. Therefore, we proceed with the variance presented in Eq 6.

Finally, as described previously, we can find the comorbidity P-values using the normal approximation of the Poisson binomial distribution. Health condition pairs with Bonferroni-adjusted $P < 0.001$ were included in the final network. Comorbid pairs were also verified by finding the Pearson correlation coefficient for the individualized health condition probabilities for each pair. Only pairs with positive coefficients were included.

## Reported onset dates and directed networks

We extended our analysis to examine temporal relationships between conditions by analyzing the owner-reported dates when each condition first occurred in their dog. For each pair of health conditions $y, z$, we analyzed the temporal relationship using only dogs reported to have both conditions, comparing their individual dates of reported condition onset to determine which condition manifested first. Unlike population-level approaches that compare average age of condition onset across all cases, our method examines the sequence of reported conditions within individual dogs' health histories.

This tells us whether a health condition is more likely to occur before another health condition. The probability that the reported onset of health condition $y$ occurs before the reported onset of health condition $z$ in a dog that has both conditions $y, z$ is

$$P_{y \to z}^{\{y,z\} \subset d} = \frac{P_{y \in d}}{P_{y \in d} + P_{z \in d}}.$$

(7)

We also incorporated a time window size $W$ and the length of the dog's medical history as done in Lemmon *et al*. [18]. Now, the probability that $y, z$ occur within $W$ given that the canine already has the two health conditions is:

$$P_{\{y,z\} \subset d \text{ in } W} = \frac{W}{Span} \left(2 - \frac{W}{Span}\right) \text{ if } Span > W,$$

$$P_{\{y,z\} \subset d \text{ in } W} = 1 \text{ if otherwise.}$$

(8)

We combined these two probabilities to get the probability that $y$ occurs within $W$ before $z$ as:

$$P_{y \to z \text{ in } W}^{\{y,z\} \subset d} = P_{y \to z}^{\{y,z\} \subset d} \times P_{\{y,z\} \subset d \text{ in } W}.$$

(9)

Finally, as described previously, we can find the direction P-values using the normal approximation of the Poisson binomial distribution. Note that for each health condition pair $y, z$, we calculated both $P_{y \to z \text{ in } W}^{\{y,z\} \subset d}$ and $P_{z \to y \text{ in } W}^{\{y,z\} \subset d}$. Health condition pairs with Bonferroni adjusted P<0.01 were included in the final network.

In our analysis we chose a window of 12 months. Other window sizes were considered, though the results are similar.

### Comorbidity network with stratification

We stratified the DAP dogs into four life-stage cohorts (from youngest to oldest): puppy, young adult, mature adult, and senior. These life stages were constructed using DAP veterinary guidelines that incorporate weight and breed in addition to age (S6 Table) [69]. Within each stratum, we selected health conditions that at least 60 dogs had and included only dogs with at least one of those selected health conditions, similar to the criteria for the unstratified network. We again used the PBC method described previously to identify potential comorbidity relationships within each age stratum. Health condition pairs with Bonferroni adjusted P<0.01 were included in the final networks, with one network for each stratum.

### Statistical Analysis

All statistical analyses were performed using R version 4.5.0 (2025-04-11) in RStudio. We used the *tidyverse* package for data manipulation and visualization, and *igraph* for network visualization. Additional specialized packages included *fastDummies* for creating dummy variables, *furrr* for parallel processing, and *poweRlaw* for degree distribution analysis. Bonferroni correction was applied for multiple testing throughout our analyses.

### Supporting information

**S1 Table. Health condition classifications and frequencies.** (A) Distribution of unique health conditions across body system categories, showing the number of distinct conditions within each category. (B) Detailed listing of all health conditions included in the study, standardized condition names, body system classifications, and frequency of occurrence in the study cohort. (XLSX)

**S2 Table. Health condition pair associations.** (A) Complete results from the unstratified, undirected comorbidity network analysis. (B-D) Results from age-stratified analyses for Young Adult, Mature Adult, and Senior dogs respectively. There is no table for the puppy cohort as no comorbid pairs were identified. (E) Results from directed network analysis showing temporal relationships between conditions occurring within a 12-month window. For all analyses, health condition pairs are shown with their names, corresponding body system classifications, test statistic, expected values and variance under the null, and p-values (unadjusted and adjusted).
(XLSX)

**S3 Table. Health condition pairs uniquely identified in age-stratified networks.** Health condition pairs that were significant in Mature Adult (A) and Senior (B) but not in any other age-stratified network. The "In Unstratified Network" column indicates whether the pair was also detected in the unstratified network analysis. Each pair is shown with condition names, numerical codes, and the age stratum where the unique association was detected.
(XLSX)

**S4 Table. Time-directed associations supported by previous literature.** Significant health condition pairs identified in our network analysis that are supported by existing veterinary literature. Each pair is shown with condition codes, names, unadjusted and Bonferroni-adjusted P-values from our analysis, and corresponding literature citations.
(CSV)

**S5 Table. Demographic characteristics of dog owners in study cohort compared to US population.** Racial distribution, educational attainment, and household income of dog owners enrolled in the study compared to 2021 US Census data. Note that owners could select multiple racial categories, so percentages sum to more than 100%. Income data from owners was collected in categorical ranges.
(XLSX)

**S6 Table. Life stage definitions across age ranges and weight classes.** The table presents the age ranges (in years) used to categorize dogs into life stages (puppy, young adult, mature adult, and senior) based on their weight class. These life stage definitions were adapted from the Dog Aging Project veterinary guidelines [69].
(CSV)

**S1 Fig. Log scale density distribution of node degrees in the undirected comorbidity network.** The overlaid lines represent the density distribution predicted by an exponential and scale-free power-law model.
(TIFF)

## Acknowledgments

The Dog Aging Project thanks study participants, their dogs, and community veterinarians for their important contributions.

## Author contributions

**Conceptualization:** Kate E Creevy, Daniel E.L. Promislow, Jing Ma.

**Data curation:** Lakshin Kumar, Kate E Creevy.

**Formal analysis:** Antoinette Fang, Jing Ma.

**Funding acquisition:** Daniel E.L. Promislow.

**Methodology:** Jing Ma.

**Supervision:** Jing Ma.

**Visualization:** Antoinette Fang.

**Writing – original draft:** Antoinette Fang.

**Writing – review & editing:** Kate E Creevy, Daniel E.L. Promislow, Jing Ma.

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
