## [Decision Letter · Decision Letter 0]

PCOMPBIOL-D-24-02174

The first comorbidity networks in companion dogs in the Dog Aging Project

PLOS Computational Biology

Dear Dr. Ma,

Thank you for submitting your manuscript to PLOS Computational Biology. After careful consideration, we feel that it has merit but does not fully meet PLOS Computational Biology's publication criteria as it currently stands. Therefore, we invite you to submit a revised version of the manuscript that addresses the points raised during the review process.

Please submit your revised manuscript within 30 days Jun 06 2025 11:59PM. If you will need more time than this to complete your revisions, please reply to this message or contact the journal office at ploscompbiol@plos.org. Please include the following items when submitting your revised manuscript:

We look forward to receiving your revised manuscript.

Kind regards,

Benjamin Hall, DPhil

Academic Editor

PLOS Computational Biology

Benjamin Althouse

Section Editor

PLOS Computational Biology

**Additional Editor Comments :**

Please address reviewers comments in full. In particular please ensure that the comments from reviewer 2 are completely addressed- in particular checking that comments referring to data collection in the study as "longitudinal" are removed to better reflect the nature of the study its limitations, noting that as a retrospective cross-sectional study it is prone to survival bias and this is not presently discussed.

All links to external datasets (i.e. github) should also be checked and where possible moved to more appropriate platforms (e.g. zenodo, which allows specific releases associated with this manuscript to be stored) for publication.

**Journal Requirements:**

At this stage, the following Authors/Authors require contributions: Antoinette Fang, Lakshin Kumar, Kate E Creevy, Daniel E.L. Promislow, and Jing Ma. Please ensure that the full contributions of each author are acknowledged in the "Add/Edit/Remove Authors" section of our submission form.

4) Your manuscript's sections are not in the correct order.  Please amend to the following order: Abstract, Introduction, Results, Discussion, and Methods

5) Please upload all main figures as separate Figure files in .tif or .eps format. For more information about how to convert and format your figure files please see our guidelines: 

6) Please upload a copy of Figure 4 which you refer to in your text on page 22 line 452. Or, if the figure is no longer to be included as part of the submission please remove all reference to it within the text.

7) Thank you for stating "The code for reproducing the results described in this study can be found at "https://github.com/aeyfang/pbc-dap-network/." This link reaches a 404 error page. Please amend this to a new link.

8) For studies involving third-party data, we encourage authors to share any data specific to their analyses that they can legally distribute. PLOS recognizes, however, that authors may be using third-party data they do not have the rights to share. When third-party data cannot be publicly shared, authors must provide all information necessary for interested researchers to apply to gain access to the data. For more information, see: https://journals.plos.org/ploscompbiol/s/data-availability#loc-acceptable-data-access-restrictions

9) Please amend your detailed Financial Disclosure statement. This is published with the article. It must therefore be completed in full sentences and contain the exact wording you wish to be published.

2) If the funders had no role in your study, please state: "The funders had no role in study design, data collection and analysis, decision to publish, or preparation of the manuscript."

10) Please ensure that the funders and grant numbers match between the Financial Disclosure field and the Funding Information tab in your submission form. Note that the funders must be provided in the same order in both places as well. Currently, these funders "Glenn Foundation for Medical Research, the Tiny Foundation Fund at Myriad Canada, the WoodNext Foundation, and the Dog Aging Institute" are missing from the Funding Information tab.

**Reviewers' comments:**

Reviewer's Responses to Questions

Reviewer #1: Thank you for this very interesting and well written manuscript. This paper describes in detail a methodology to identify comorbidities in canine patients through construction of comorbidity networks, using data obtained from the Dog Aging Project. This will potentially be of great help to clinicians in reaching diagnoses and anticipating concurrent conditions in their patients. It is also likely to be of interest to the public, especially pet owners who might recognise the comorbidities identified here in their own animals. The methodology appears sound and replicable, although I write this as a clinician-epidemiologist, so would recommend the editor also takes advice on this from a colleague in a suitable mathematical field as necessary.

I have some minor comments which I hope you will find helpful.

1) Title: This paper focuses strongly on methodology rather than clinical application, and much of the information of interest to clinicians is relegated to the supplementary material. I feel that the title does not make clear this primary focus on methodology in its current form.

2) Line 74: Please could you explain the reason for choosing a cutoff of 60 dogs?

3) Line 80: Use of lbs is not standard in many countries, and the Plos Comp Biol author guidelines advise use of SI units. Please could SI units be added in parentheses throughout where imperial units have been used?

4) Fig. 1, 2, and 3: These figures are not interpretable from a clinical perspective without supplementary table S1b. I wonder if this table needs to be included in the main body of the paper rather than as a supplement.

5) Line 194: Fig S4 shows 13 mature adult and 1 senior comorbidity pairs, not ten and two?

6) Line 283: How is age calculated? Does this relate to the onset of the condition, or start of treatment, or another date?

7) Line 283: How is weight calculated? How are multiple episodes of weight recording in an individual patient record taken into account?

8) Line 383: Should this reference Fig. S4?

9) Line 452: There is no Fig 4?

10) Line 501: The GitHub link is not currently active.

Reviewer #2: Summary

This paper entitled 'The first comorbidity networks in companion dogs in the Dog Aging Project' was a very interesting read and I really enjoyed it. The authors use mathematical approaches to understand comorbidities in dogs, which have rarely been applied within veterinary epidemiology. In addition to using these novel approaches, the paper provides useful insights into dog ageing and health. The paper is nicely written and structured, with excellent grammar and no typing errors. I have a few comments about the reporting clarity, which at times makes it a challenge to understand the methods and results. At times, it is not clear why some results have been reported in the main text of the manuscript and discussed, and not others, which seem equally important.

I take grievance to the authors referring to the data collection used in this study as ‘technically cross-sectional’ but able to ‘capture lifetime prevalence’. Although the long-term plan is for a longitudinal cohort, the Dog Aging Study at this point of data collection is a retrospective cross-sectional study and the authors should not try to infer otherwise (the study is referred to as longitudinal several times), which is confusing to the reader and ignores recall bias (although recall bias is later discussed). This ignores the fact that this type of study design is prone to survival bias, which is a major limitation that has been well-documented across many epidemiological studies, but isn’t included as a limitation in the discussion.

Introduction

Page 2, lines 49-50 “Notably, dogs are a good model organism because they share the same environments and many of the same health conditions as humans, they have an almost equally sophisticated healthcare system, and show similar manifestations of aging as humans” - I would suggest that dogs are in fact, treated much differently than humans - the onus in humans is to keep them alive as long as possible, whereas in veterinary healthcare it is far more about economic cost to the owner and also the moral/ethical considerations to the animal

Results

Although I like Figures 1, 2 and 3, it is impossible to understand the results using those figures alone, without referring to the supplementary material. Because of the magnitude of the results, it seems that many findings are easy to ‘lose’ and the author’s seem to have picked out certain results but do not explain many counter-intuitive ones. I wonder if some of the main results could be brought forward into the main manuscript to improve the reader experience and overall trust in what the author’s have chosen to discuss..

Page 4 lines 83-84 “Additionally, most dogs have three or fewer health conditions, with just 37% of dogs having four or more health conditions” this seems like quite a large proportion to be phrased as ‘just’, perhaps rephrase? Additionally, these dogs technically have multimorbidity not comorbidity!

Page 4 Table 1 - please report percentages to 2 decimal places

Page 5 Table S2 - Cancers seem to have all important correlations with age and weight, with extremely low P-values reported across them all. This result seems to stand out to me, so why is it not reported?

I understand that not every result can be reported in the text here due to the magnitude of the results, but I would recommend that as a minimum, where results are reported in the text, that this is then discussed further in the discussion. Otherwise, it is not clear why these results were reported and not others. For example:

Page 6 lines 107-109 “Purebred dogs showed significantly higher rates of ear infection, cataracts, and intervertebral disk disease.” This is not mentioned in the discussion - can we link to other research that shows similar findings?

Page 6 lines 108-109 “In comparison, mixed-breed dogs had significantly higher rates of cruciate ligament rupture, being hit by a car, pruritis, and hookworms.” - There is no further discussion of these results. Could this be that owners who own mixed breed dogs are less likely to worm them regularly or allow their dogs to roam freely more often? This is where owner-demographic data would have been a nice addition to this study, but I am not sure if this was possible.

Page 10 lines 182-183 “This likely reflects the relatively low frequency of multiple health conditions in young dogs rather than simply a statistical power limitation” - this is a fairly big claim to make, did you do the power limitation calculation?

Page 10 lines 194-195 “Ten of these pairs came from the mature adult group and two from the senior group (Fig. S4).”. Some of these are intuitive e.g lameness & osteoarthritis, but several are not - ticks and allergies, cataracts and deafness, conjunctivitis and giardia. There is no elaboration/discussion about these counter-intuitive associations and it does make me worry that some of the results feel ‘cherry-picked’.

Page 12 lines 226- 227 “we examined temporal relationships between conditions by computing the probability of one condition preceding another within a time window of 12 months” - in several places throughout the paper this time window of 12 months is referred to and even in the discussion there is mention of trying other time windows, but there is no justification of why the authors chose this time window of 12 months. Can you clarify? Were other time windows tested?

Fig S1 is very useful for understanding the results and I would consider including it in the main manuscript. However, why is “chocolate” one of the most frequent diseases? This needs some explanation and perhaps adding the disease category it was assigned to (toxic consumption) would be useful (in addition to the other comorbidities)..

Methods

Include a brief description of how you completed the analysis - in R software? What version? What packages were used? Where is the code available? What are your reporting statistics?

Why were confidence intervals not reported? This would be useful for the reader.

Data cleaning is a prominent issue in questionnaire data - can you summarise how data were cleaned and processed?

Page 14/15 - lines 268-270 “As such, while the data we analyze are technically cross-sectional in nature because they were reported at a single point in time, the survey items are constructed to capture lifetime prevalence as defined above.” They are not ‘technically’ cross-sectional - they are cross-sectional. It is a retrospective cross-sectional study. Please see my other comments about this and survivor bias.

Page 15, lines 272-273 “Of note, the following information from this dataset was used: (1) age, sex, sterilization status, breed background, and weight of each dog”. Why were the following variables decided as confounders - age, sex, sterilization status, breed background, and weight of each dog, are there no other variables of relevance in the DAP?

Page 15, lines 285-286. “We used breed background as opposed to breeds as a covariate because there are a large number of breeds and many health conditions are very rare.” How sure are you on breed background - are dogs registered (e.g with the Kennel Club or other breed associations) or is this just owner reported? Have they been genetically tested or is it not possible to actually ascertain whether the dogs are pure-bred or not?

Page 16 lines 264-266 “Owners are presented with more than 300 different health conditions or clinical signs, with an option for “other, please describe” in each body system, and also provide the time of onset for each reported condition” Was any effort made to identify/classify the diseases in the other category? It is not clear how this data was used as it is not reported in the results?

Page 18 line 58. Fig S3 is very useful for understanding the results, I would consider including it in the main manuscript, with overlaid cutoffs of the age categories that you defined explicitly to show young, mid and senior dogs - see my other comment about age categories.

Page 19 lines 376-377 “ We stratified the DAP dogs into four life-stage cohorts (from youngest to oldest): puppy, young adult, mature adult, and senior. These life stages were constructed using DAP veterinary guidelines” - State explicitly what the cut-offs were for these age categories. This will make the results easier to compare to other findings and also to understand how these figures relate to the information in Figure S3.

Page lines 459 - 462 “at least at the time of enrollment, dogs in the Dog Aging Project might be healthier than average if owners are disinclined to sign up a very unhealthy dog for a study on the determinants of healthy aging, similar to the well-known “healthy volunteer” effect in human studies”. I agree, but this should not be understated. In addition to healthy volunteer bias, survival bias is completely ignored here and is perhaps the most important limitation of this study. To elaborate, this sample comprises ONLY dogs who have survived to the point of entry into the study, (and no dogs that are already dead are included). This is a major limitation of the study design as it can distort the true relationships between comorbidities. This is reflected in the fact that there are few ‘fatal’ diseases reported in this study. For example, what about the dogs who died of cancer? I would like to see at least a couple of sentences discussing survivor bias, as it has huge implications for your findings.

Page 21 lines 420-422 = “ For instance, this time-directed network exhibits a cluster of parasite-parasite interactions. While interest in co- or multiparasitism has increased in recent years, more work is needed to understand the mechanisms of these interactions” - These results are highlighted several times in the paper but I don’t think it is one of the more interesting findings. It is not surprising that several parasites are occurring together if owners simply are not worming their dogs?

Page 21 lines 451 and 452 - “This issue likely explains some counterintuitive results like tick-borne health conditions preceding ticks in the time-directed network” - although the results are available in the figure cited, you have not reported this in the results text. Due to issues with the interpretability of the figures in their standalone form (see my comment at the start of results), this was not obvious when reading the results.

Page 23, lines 462-463. “the owners in our study cohort are predominantly white, highly educated, and wealthier than the US population” It would be interesting to investigate the impact of this in future work.

Finally, this paper focuses on comorbidity and pairs of diseases, but multimorbidity is of huge topical importance in the field of human research right now and it would be an interest area to develop and expand this work (e.g., using network analyses), so might be worth a mention!

Reviewer #3: Dear authors, the following are suggestions and comments:

- Table 1 is not necessary because it does not represent the sample. The data contained therein can be presented in text only.

- Unlike the data previously cited, the data in lines 90-112 could be better visualized and didactically presented by groups in a table, with real statistical confirmations.

- I understand that Figure 1 is a complex figure, however it is an element that fails to present the data well because it is not self-explanatory. The use of acronyms and codes that are not described is a fundamental factor in this lack of clarity. I know that the authors may have a tendency to explain the figure as complex and with codes that are well known somewhere, however an incomprehensible graphic element is of no scientific use. I strongly recommend that the authors rethink Figure 1 so that it is didactic and actually demonstrates something, or remove it from the text (which is not the most recommended). Describe the legend in much more detail to facilitate this construction. Referring to Table S1 is also not a convenient idea.

- The same previous comment applies to Figures 2B, 2C and 3.

- I really missed a robust conclusion about the data discussed. The data is robust, the analysis is also, and very rich in information, where the authors, in the end, did not even elaborate a conclusion with real applications on the data presented.

**Have the authors made all data and (if applicable) computational code underlying the findings in their manuscript fully available?**

Reviewer #1: Yes

Reviewer #2: Yes

Reviewer #3: Yes

PLOS authors have the option to publish the peer review history of their article (what does this mean? ). If published, this will include your full peer review and any attached files.

**Do you want your identity to be public for this peer review?** For information about this choice, including consent withdrawal, please see our Privacy Policy .

Reviewer #1: No

Reviewer #2: No

Reviewer #3: No

**Figure resubmission:**
---

## [Editor Report · Decision Letter 1]

Dear Dr. Ma,

We are pleased to inform you that your manuscript 'Constructing the first comorbidity networks in companion dogs in the Dog Aging Project' has been provisionally accepted for publication in PLOS Computational Biology.

Best regards,

Benjamin Hall, DPhil

Academic Editor

PLOS Computational Biology

Benjamin Althouse

Section Editor

PLOS Computational Biology

Thank you for your resubmission. I am happy to recommend acceptance of the manuscript following your edits and responses to reviewers.

I would strongly recommend that in the copy editing process you add a license to your code, and move it to a repository that allows a DOI to be associated with this specific version used here e.g. Zenodo, following the journal's recommendations about code availability https://journals.plos.org/ploscompbiol/s/code-availability

---

## [Editor Report · Acceptance letter]

PCOMPBIOL-D-24-02174R1

Constructing the first comorbidity networks in companion dogs in the Dog Aging Project

Dear Dr Ma,

I am pleased to inform you that your manuscript has been formally accepted for publication in PLOS Computational Biology. Your manuscript is now with our production department and you will be notified of the publication date in due course.

With kind regards,

Zsofia Freund
